civil engineering/graph theory

origin–destination network, urban hotspots, COVID-19, movement patterns, epidemical control, census-block-group to point-of-interest movement network

**Author for correspondence:**
Qingchun Li
e-mail: qingchunlea@tamu.edu

# Disparate patterns of movements and visits to points of interest located in urban hotspots across US metropolitan cities during COVID-19

Qingchun Li[1], Liam Bessell[2], Xin Xiao[2], Chao Fan[1], Xinyu Gao[1] and Ali Mostafavi[1]

[1]Zachry Department of Civil and Environmental Engineering, and [2]Department of Computer Science and Engineering, Texas A&M University, 199 Spence Street, College Station, TX 77843-3112, USA

QL, 0000-0003-0769-0238; LB, 0000-0003-1777-7812

We examined the effect of social distancing on changes in visits to urban hotspot points of interest. In a pandemic situation, urban hotspots could be potential superspreader areas as visits to urban hotspots can increase the risk of contact and transmission of a disease among a population. We mapped census-block-group to point-of-interest (POI) movement networks in 16 cities in the United States. We adopted a modified coarse-grain approach to examine patterns of visits to POIs among hotspots and non-hotspots from January to May 2020. Also, we conducted chi-square tests to identify POIs with significant flux-in changes during the analysis period. The results showed disparate patterns across cities in terms of reduction in hotspot POI visitors. Sixteen cities were divided into two categories using a time series clustering method. In one category, which includes the cities of San Francisco, Seattle and Chicago, we observed a considerable decrease in hotspot POI visitors, while in another category, including the cities of Austin, Houston and San Diego, the visitors to hotspots did not greatly decrease. While all the cities exhibited overall decreased visitors to POIs, one category maintained the proportion of visitors to hotspot POIs. The proportion of visitors to some POIs (e.g. restaurants) remained stable during the social distancing period, while some POIs had an increased proportion of visitors (e.g. grocery stores). We also identified POIs with

significant flux-in changes, indicating that related businesses were greatly affected by social distancing. The study was limited to 16 metropolitan cities in the United States. The proposed methodology could be applied to digital trace data in other cities and countries to study the patterns of movements to POIs during the COVID-19 pandemic.

## 1. Introduction

The objective of this study is to examine movement patterns to urban hotspots in United States cities during the initial 2020 COVID-19 outbreak. Urban mobility and movement patterns are important characteristics of urban dynamics, reflecting the collective human behaviour and social interactions [1,2]. Urban mobility drives the spatial flux of populations, and effective epidemic control measures greatly rely on the characterization of urban mobility patterns [3–7]. Assessment of urban mobility is an important element of epidemic control [8]. Most standard epidemic models employ mobility patterns in prediction of a disease outbreak trajectory. Tizzoni *et al.* used commuter movement data to model the spatial spread of epidemics in European countries [9]. The study examined whether the mobility data matched the empirical mobility patterns and how the observed discrepancies of mobility patterns would affect the results of influenza-like illnesses spread simulation. Balcan *et al.* [10] developed a worldwide epidemic model to evaluate the force of infection based on the description of mobility patterns obtained by the gravity model. The results showed that long-range airline traffic determined the global epidemic dynamic, while the short-range mobility patterns determined the local epidemic diffusion pattern. Ferguson *et al.* [11] developed a transmission model for H5N1 influenza in Southeast Asia taking the community mobility into consideration. The model tested containment strategies such as prophylaxis and social distancing measures under different reproduction number of the virus. Meloni *et al.* [12] found that it was essential to consider how the epidemic directives enacted by states, for instance, induced changes in mobility patterns and how the changes in turn affected the propagation of the epidemic. Meloni *et al.* [12] developed an epidemic model accounting for changes in mobility patterns due to the response to an epidemic outbreak. The results showed that self-initiated behavioural changes (e.g. changes in travelling routes) may accelerate the spread. These studies and models highlight the necessity of characterizing mobility and movement patterns for better understanding the extent and trajectories of COVID-19 in metropolitan urban areas.

While the reduction in overall movements and mobility could promote containment, it is equally important to monitor and evaluate movement reduction to urban hotspots. In comparing the effectiveness of social distancing measures between cities, it has been observed that epidemic spread trajectories are different, while the overall mobility reduction is similar across cities. These disparate trajectories could be in part due to differences in movement patterns to urban hotspots. Urban hotspots and sub-centres usually have higher populations and employment densities and more points of interest (POIs) compared with other areas of cities [13,14]. Urban hotspots and sub-centres, therefore, are gravity activity centres affecting population movement, mobility patterns and human interactions. In a pandemic situation, however, urban hotspots could be potential 'superspreader' POIs [15], because visits to hotspots can greatly increase the risk of contact and transmission of disease. Understanding mobility patterns of the visiting of urban hotspots is important for developing and monitoring effective epidemic control measures. Origin–destination (OD) network analysis provides a powerful tool to study mobility patterns under such a situation and are especially helpful for locating hotspots and studying the urban mobility patterns of visiting urban hotspots [16,17]. Louail *et al.* and Hamedmoghadam *et al.* used the OD matrix and a coarse-grain approach to study the mobility among hotspots and non-hotspots [1,18,19]. The OD matrices aggregate the mobility of individuals from one point to another [20,21]. Therefore, the OD matrices include insightful information of population movements and patterns of movements within and across cities [19,22]. In addition to traditional surveys and counting to develop OD matrices, increasing studies extracted OD matrices based on the emerging digital footprint data [23,24]. Mazzoli *et al.* [22] extracted the OD matrices from Twitter data to map daily commuting flows in London and Paris. Lenormand *et al.* [25] mapped the OD matrices from three datasets, including Twitter, mobile phone and census data. This study showed strong correlations between three datasets regarding individual mobility patterns, lending support to interchanging the three datasets to study mobility patterns.

In summary, the extant studies demonstrated that the characterization of urban mobility and movement patterns are important to understand the collective human behaviour and social

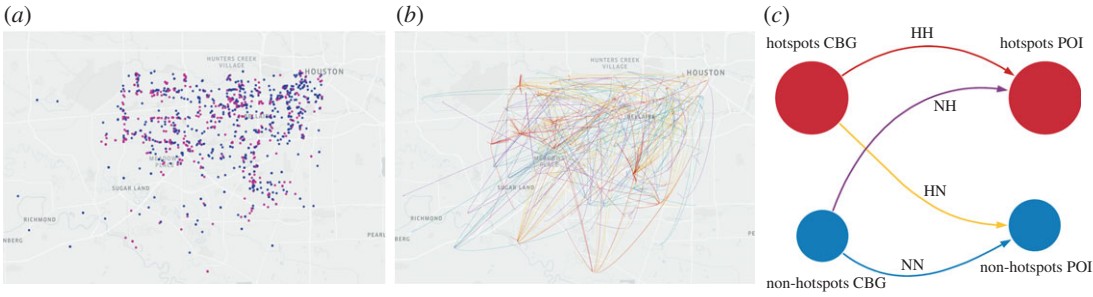

**Figure 1.** The coarse-grain approach categorizes CBG-POI movements to four types of movements among hotspots and non-hotspots: (*a*) hotspots (red nodes) and non-hotspots (blue nodes), (*b*) individual CBG-POI movements among hotspots and non-hotspots, four colours represent four types of movements, (*c*) clarification of four types of movements among hotspots and non-hotspots. (Figure 1 was plotted based on the SafeGraph data for Houston.)

interactions, which are critical for the development of effective epidemic control measures. Also, urban hotspots are gravity activity centres that usually have a higher density of populations and POIs, which could be potential 'superspreaders' during the pandemic situation. Therefore, understanding the patterns of visits to POIs in urban hotspots is important for developing and monitoring epidemic control measures. Extant studies, however, rarely studied the patterns of visits to POIs in urban hotspots under pandemic situations. Hence, in this paper, we investigated the patterns of population visits to urban hotspots using origin–destination networks from census-block-groups (CBGs) to POIs in 16 cities of United States based on the digital trace data from SafeGraph. The POI data enable the identification of urban hotspots to evaluate changes in visiting urban hotspots due to social distancing measures during the COVID-19 pandemic. We also identified POIs that had significant flux-in decreases and what POI-associated business categories were greatly affected during COVID-19. These POIs and business categories could expect a significant flux-in increase after the shelter-in-place orders are lifted. The results of this study could help decision-makers better monitor and evaluate epidemic/pandemic control measures, as well as reopening policies and strategies. In this paper, we adopted a modified coarse-grain approach for separating the hotspot and non-hotspot nodes in mapped CBG-POI movement networks [1]. Hotspot and non-hotspot nodes are computationally determined. Hotspots are nodes with higher weighted degree centrality, and non-hotspots are nodes with lower weighted degree centrality in the mapped CBG-POI networks. The adopted method determined the threshold to separate the hotspot and non-hotspot nodes. A detailed explanation of the adopted approach is presented in the Data and methodology part. Figure 1 illustrates four types of movements among hotspots and non-hotspots.

## 2. Data and methodology

We used POI data provided by SafeGraph to map the CBG-POI movement networks. SafeGraph aggregates POI data from diverse sources (e.g. third-party data partners, such as mobile application developers) and removes private identity information to anonymize the data. The POI data include base information of a POI, such as the location name, address, latitude, longitude, brand and business category. SafeGraph uses the standard North American Industry Classification System (NAICS) to classify POI business categories. The data reveal the visit pattern of POIs including the aggregated number of visits to the POI during the data range, the number of visits to the POI each day over the period and the aggregated number of visitors to the POI from CBGs during the period (e.g. one week and one month).

In this paper, we used the POI data: Weekly Pattern Version 2, to study movement patterns in 16 cities in the United States. The Safegraph weekly pattern data provide information related to the visits to POIs and cover the entire United States. The data were aggregated weekly (Monday to Sunday) [26]. Among these 16 cities are 14 largest cities in the United States by population. We selected the top 14 largest cities in the United States due to two considerations: (i) Safegraph collected more data in larger cities. We tested several less-populated cities in the United States such as Randolph, Terrell and Early in Georgia, as well as Union, Bergen and Hudson in New Jersey. The visitors to POI data in these cities, however, were sparse, and it was infeasible to build the CBG-POI movement networks,

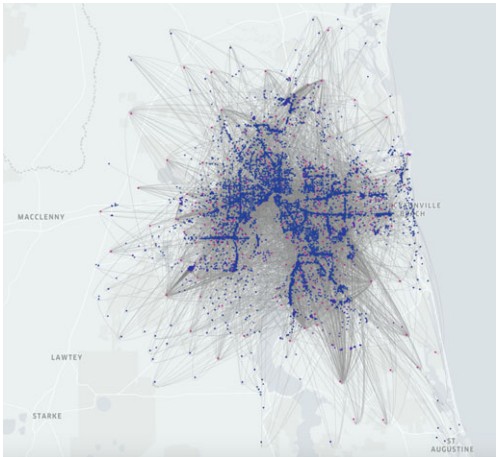

**Figure 2.** Mapped CBG-POI movement network for the week of 27 January 2020, in Jacksonville, Florida. The figure shows a total of 83 661 weighted edges. Red nodes represent hotspots (1314 nodes) and blue nodes represent non-hotspots (10 820 nodes).

and (ii) we were more concerned about the spread of COVID-19 in cities with a larger population. In addition, Seattle and Detroit were studied. Seattle was the first city in the United States to report a diagnosed COVID case, and Detroit had a burst in the number of confirmed cases in March 2020. Also, we considered that Detroit is the largest city in the midwestern state of Michigan, and the number of confirmed cases in Detroit greatly decreased after April 2020. Therefore, Detroit became a good example to compare with other cities. The analysis comprises four major steps: (i) map the CBG-POI movement network, (ii) identify hotspots and non-hotspots based on the mapped network, (iii) examine movement patterns between hotspots and non-hotspots, and (iv) identify POIs with significant flux-in changes. We explain each step in the following sections.

## 2.1. Map the origin–destination network

We mapped the CBG-POI movement networks based on the number of visitors to POIs from CBGs. The mapped CBG-POI networks are directed and weighted bipartite networks, where pairs of nodes $i$ and $j$ represent CBGs and POIs with mapped geolocation. Links in the CBG-POI movement networks represent visits from CBGs to POIs, and non-negative weights of links, $w_{ij} > 0$, represent one or more visitors during the covered period. If there was no movement from CBGs to POIs, $w_{ij} = 0$. We mapped the weekly CBG-POI movement network because SafeGraph aggregates the number of visitors from CBGs to POIs by week. Figure 2 illustrates an example of the mapped CBG-POI movement network in Jacksonville, Florida.

## 2.2. Identify hotspots and non-hotspots

In this study, we focused on the patterns of visits to POIs from CBGs to examine movement patterns across hotspot and non-hotspot clusters. Therefore, the coarse-grain approach that clusters hotspot and non-hotspot nodes in CBG-POI movement networks were used [1,18,19]. In the existing literature, different methods have been proposed to separate hotspots and non-hotspots. Louail *et al*. [18,19] developed a method to separate hotspots and non-hotspots based on the Lorenz curve of divided $1 \, km^2$ cells. This method yields lower and upper boundaries of hotspots. Hamedmoghadam *et al*. [1] showed that using Lorenz curve to identify hotspots and non-hotspots was biased to the outlier nodes. Therefore, they proposed a modified coarse-grain approach using a centroid-based clustering method to separate hotspots and non-hotspots. In this paper, we adopted the modified coarse-grain approach that separated hotspot and non-hotspot nodes in the mapped CBG-POI movement networks. The employed method determined hotspot and non-hotspot nodes in CBGs and POIs separately. Therefore, different spatial resolutions of CBGs and POIs will not affect the results of hotspot and non-hotspot nodes in CBGs and POIs. Each mapped weekly CBG-POI network has a correspondent CBG-POI bi-adjacency matrix. The columns and rows of the CBG-POI matrix represent origin nodes and destination nodes, and the elements are the weights of links. First, we summed all the rows and columns to get the total flux-out and flux-in values of CBG and POI nodes, respectively.

Then, we sorted flux-out and flux-in values of CBG and POI nodes in an ascending order: $O_1 < O_2 < \cdots < O_n$ and $D_1 < D_2 < \cdots < D_n$. To separate hotspots and non-hotspots in these two lists, we used equation (2.1) to determine the separation point $O_c$ and $D_c$. Nodes with flux-out and flux-in values greater than $O_c$ and $D_c$ are hotspots of CBGs and POIs. In equation (2.1), $q_i$ could represent either $O_1, O_2, \cdots, O_n$ or $D_1, D_2, \cdots, D_n$.

$$\arg\min_c \sum_{i=1}^{c} \left| q_i - \frac{1}{c}\left(\sum_{k=1}^{c} q_k\right) \right| + \sum_{j=c+1}^{n} \left| q_j - \frac{1}{n-c}\left(\sum_{l=c+1}^{n} q_l\right) \right|. \tag{2.1}$$

## 2.3. Examine movement patterns between hotspots and non-hotspots

We used a coarse-grain approach to examine the mobility pattern [1,18,19]. The approach reduces the mobility flows to four types: (i) HH: from hotspot CBGs to hotspot POIs, (ii) NH: from non-hotspot CBGs to hotspot POIs, (iii) HN: from hotspot CBGs to non-hotspot POIs, and (iv) NN: from non-hotspot CBGs to non-hotspot POIs. If we use $F$ to represent the original CBG-POI bi-adjacency matrix, then we could reduce $F$ to the coarse-grained matrix $\Lambda$

$$\Lambda = \begin{bmatrix} \text{HH} & \text{NH} \\ \text{HN} & \text{NN} \end{bmatrix}. \tag{2.2}$$

In matrix $\Lambda$, each sub-matrix could be calculated as follows [1,18,19]:

$$\text{HH} = \frac{\sum_{i \in M, j \in p} F_{ij}}{\sum_{i,j} F_{ij}}, \tag{2.3}$$

$$\text{NH} = \frac{\sum_{i \notin M, j \in p} F_{ij}}{\sum_{i,j} F_{ij}}, \tag{2.4}$$

$$\text{HN} = \frac{\sum_{i \in M, j \notin p} F_{ij}}{\sum_{i,j} F_{ij}} \tag{2.5}$$

and

$$\text{NN} = \frac{\sum_{i \notin M, j \notin p} F_{ij}}{\sum_{i,j} F_{ij}}, \tag{2.6}$$

where $F_{ij}$ represents each element in the original CBG-POI matrix, $M$ represents the set of hotspot CBGs and $p$ represents the set of hotspot POIs determined in step 2. Equations (2.3)–(2.6) illustrate how we calculated the proportion of each type of movements. We normalized each mobility type by the total mobility flow. Therefore, the proportion of each type of movement, HH, NH, HN and NN $\in [0,1]$, and the sum of them equals 1. HH, NH, HN and NN could represent the proportion of each type of movement flow in the whole CBG-POI movement network [1]. This characterization is particularly important to examine and monitor reduction in movements to urban hotspots (reduction in the proportion on HH and NH movements) during social distancing periods.

## 2.4. Clustering analysis of movement patterns across cities

We conducted a clustering analysis after we obtained four types of movements in cities. We used the sum of proportions of two movements, HH + NH, as an indicator to cluster movement patterns within cities. These two movements would contribute to the spreading of the epidemic because the extent of visits to the hotspot POIs could increase the transmission rate of COVID-19. We scaled the time series data related to movement patterns so that each time series had zero mean and unit standard deviation. This step enabled us to focus on comparing the shapes and trends of time series data. We compared three algorithms (Euclidean distances, dynamic time warping (DTW), cross-correlation) for time series clustering [27–29] and used the silhouette coefficient to determine the number of clusters [30]. (Results of the algorithms are presented in the electronic supplementary material.)

## 2.5. Identify points-of-interest with significant flux-in changes

We compared CBG-POI matrices from two milestone dates (e.g. 1 March and 29 March). We summed columns of the matrices to obtain the weighted node degree centrality of POIs. Then we calculated differences in the weighted degree centrality of each pair of POI nodes in the two matrices:

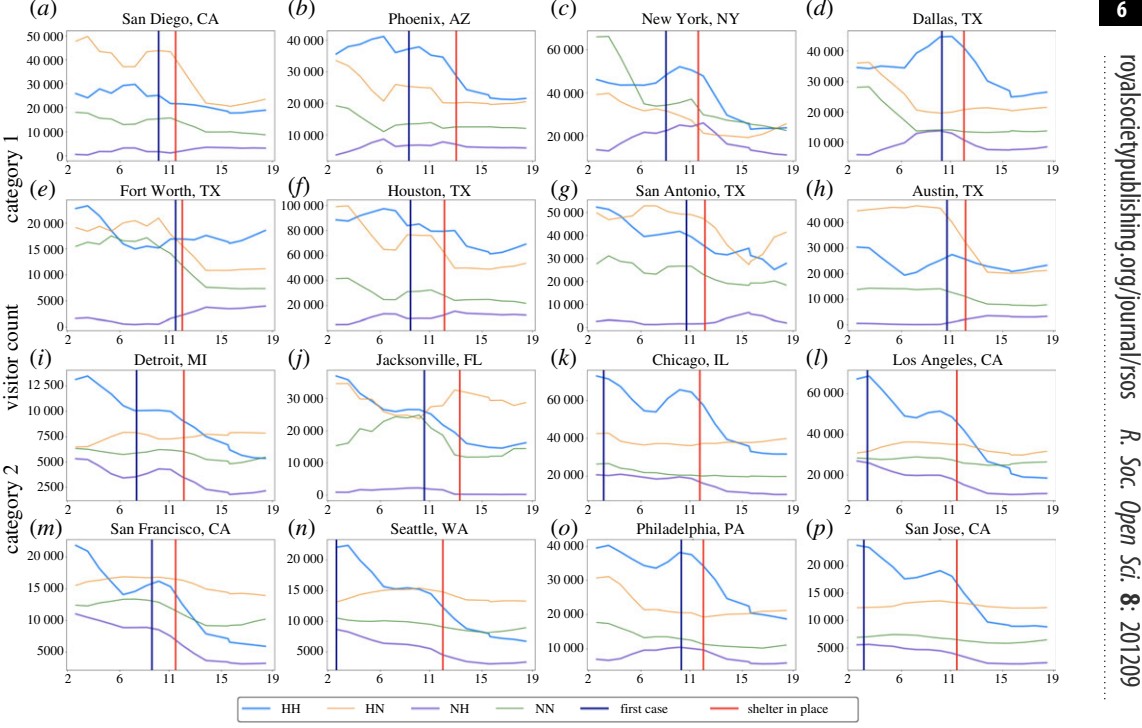

**Figure 3.** Visitors related to four types of movements in 16 cities. We used the rolling mean (window = 4) to smooth the data. For the original data, please see the electronic supplementary material.

$C_1, C_2, \cdots, C_n$. Accordingly, $C_n^2/\overline{C^2}$ will approximately follow the chi-square distribution if the weighted node degree centrality did not have significant changes. (Proof process is presented in the electronic supplementary material.) Here, $\overline{C^2}$ is the average of the square of degree centrality difference. We used the upper tail test ($H_1: |C| > 0, H_0: |C| = 0$) of the chi-square distribution to determine the $p$-value for each node. Because we conducted each test separately for each node, the degree of freedom is 1, and we adjusted $p$-value for multiple tests using the Benjamini–Hochberg false discovery rate (FDR) correction [31]. We tested the POI node set and identified the POIs with significant in-degree changes (with FDR equal to 0.1 and adjusted $p$-value < 0.01), which could reflect significant flux-in changes. Furthermore, because each POI has its NAICS code indicating its business activity, we can identify the extent to which social distancing measures affected business activities.

# 3. Results

## 3.1. Movement patterns of visiting points-of-interest in 16 cities

Figure 3 illustrates that the sum of visitors to POIs showed a decreasing trend for all 16 cities after the enforcement of shelter-in-place orders. However, four types of movements (HH, HN, HH and NH) varied across different cities. Because we only compared the increasing or decreasing trends of four types of movements among cities, the population size differences between the cities would not affect the results. Figure 3 includes the result of clustering analysis, and the 16 cities were divided into two categories. The upper eight cities are category 1 and the lower eight cities are category 2. Figure 4 shows the proportion of each type of movements in 16 cities. Data shown in figure 4 were normalized by the total number of four types of movements in each city. Figure 5 illustrates the detailed clustering results.

We can observe from figure 3 that the HH movements in category 1 cities (San Diego, Fort Worth, Dallas, Houston and Austin) did not show a clear declining trend. In category 1 cities, a decline in HN and NN movements caused a decrease in the total number of visitors. In fact, the NH movements in cities of category 1 (except for New York) showed an increasing trend after the enforcement of shelter-in-place order. In category 2 cities, HN and NN movements remained stable, while HH and NH movements show a clear downward trend.

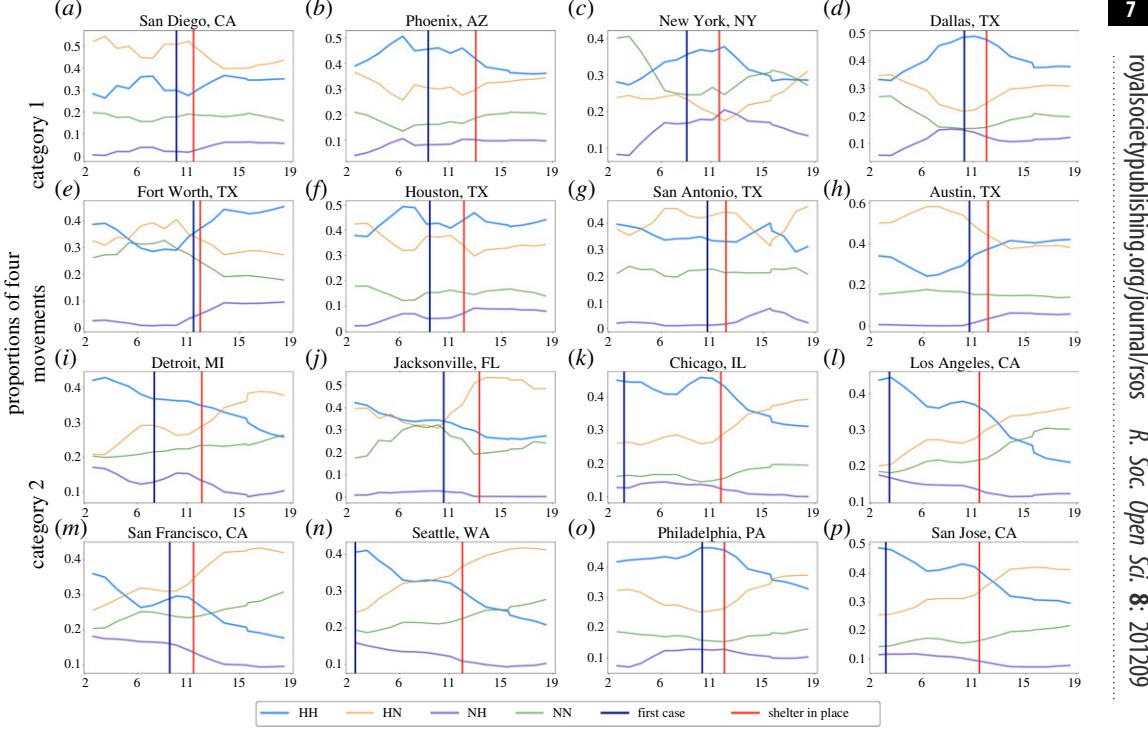

**Figure 4.** Proportions of four types of movements in 16 cities, rolling mean (window = 4).

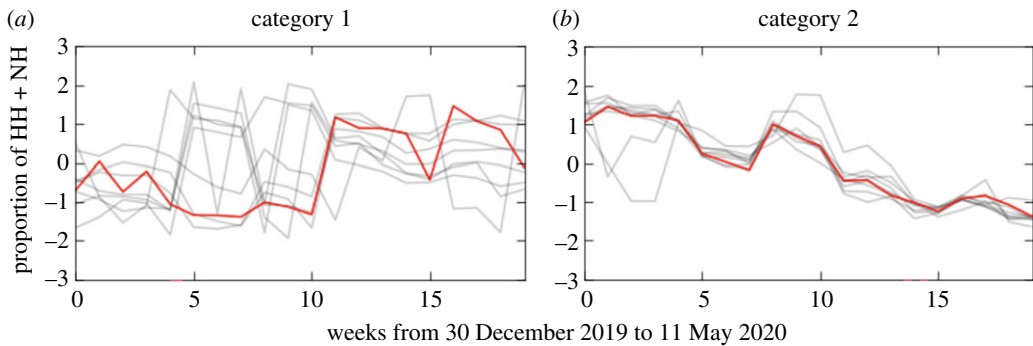

**Figure 5.** Result of clustering analysis using dynamic time warping barycentre averaging. Each grey line represents movement of one city in categories; the red line represents the barycentre of the category.

We also investigated the proportion of each type of movements in 16 cities. We can observe from figure 4 that the proportion of HH and NH movements in category 1 cities did not show a declining trend, even though the absolute value of HH and NH movements in some cities, such as Phoenix, San Antonio and New York, declined (as shown in figure 3). The proportion of HN and NN movements in most cities (except for New York) of category 1 did not show a clear upward trend. This result demonstrates that although category 1 cities had decreased absolute visitors to POIs, the proportion of their visitors to the hotspots of POIs were stable.

For category 2 cities, the proportion of HH and NH movements showed a clear downward trend, while the proportion of HN and NN movements had a clear upward trend. The barycentre of two city categories illustrated in figure 5 indicates that the proportion of HH and NH movements had an upward trend in category 1 cities, while the proportion of HH and NH movements showed a downward trend in cities of category 2. These results imply that cities of category 2 had decreased proportion of their visitors to urban hotspots of POIs due to the social distancing measure. We can conclude from the above results that while the overall mobility in all cities declined due to social distancing orders, the movement patterns related to visits to hotspots followed two different trends in the two categories of studied cities. The disparate patterns could imply differences in transmission risks.

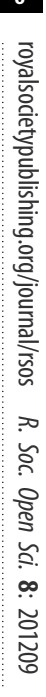

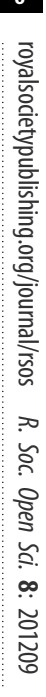

**Figure 6.** Top seven hotspot POIs in hotspots during the last week of January, February, March and April.

In addition, we can observe from figure 4 that the proportion of HH and NH movements in some category 2 cities (Detroit, Jacksonville, Chicago, Los Angeles, San Francisco, Seattle and San Jose) started to decline much earlier than the enforcement of the shelter-in-place orders. This result may imply these cities had started to proactively reduce visitors to hotpots. The first cases occurred quite early in most of these cities, such as Detroit, Chicago, Los Angeles, Seattle and San Jose. As there is a clear gap between the date of the first case and the enforcement of shelter-in-place orders in these cities, this result may suggest that the information of the first case may trigger proactive actions.

## 3.2. Proportion of persons visiting points-of-interest in hotspots

The next set of results indicates one dominant POI in hotspots across all 16 cities: restaurants and other eating places (NAICS code: 7225). Museums, historical sites and similar institutions (NAICS code: 7121) was the second dominant POI in many of the studied cities. Based on the description of NAICS, museums, historical sites and similar institutions encompass several sub-categories, including museums, historical sites, zoos and botanical gardens, nature parks and other similar institutions. Surprisingly, the proportion of visitors to these two POIs remained fairly stable during the unfolding of the COVID-19 pandemic and the enforcement of shelter-in-place orders. Figure 6 illustrates the top

seven hotspot POIs with the highest proportion of visitors in four cities: Austin, New York, San Francisco and Seattle (The results for the other cities are provided in the electronic supplementary material.) We selected two cities from each category and one week at the end of each of January, February, March and April to illustrate the patterns.

As illustrated in figure 6, the proportion of visitors to POIs in hotspots showed a similar pattern in the weeks of 27 January 2020, and 24 February 2020, in addition to the two dominant POIs (i.e. restaurants and museums), other amusement and recreation industries (NAICS code: 7139) ranked third for three cities (ranked fourth in Austin), while the fourth and fifth place POIs varied across cities: child day-care services (NAICS code: 6244) and traveller accommodation (NAICS code: 7211) in New York; sporting goods, hobby and musical instrument stores (NAICS code 4511) and traveller accommodation (NAICS code: 7211) in San Francisco; as well as sporting goods, hobby and musical instrument stores (NAICS code: 4511) and grocery stores (NAICS code: 4451) in Seattle. In Austin, lessors of real estate (NAICS code: 5311), gasoline stations (NAICS code: 4471) and elementary and secondary schools (NAICS code: 6111) had large proportions of visitors. With the unfolding of COVID-19 and shelter-in-place orders, although the proportion of visitors to the top two POIs slightly decreased, the top two POIs in each city still were the dominant places visited. After the unfolding of COVID-19 and social distancing orders, the proportion of visitors to grocery stores (the red element in figure 6) increased. Also, the proportion of visitors to other amusement and recreation industries and travel accommodation declined. For the week of 30 March, grocery stores started to rank fifth while another essential POI, gasoline stations, ranked fourth in Austin. Also, grocery stores started to rank fourth in the weeks of 30 March and 27 April in the other three cities. In Austin, the proportion of grocery stores visitors decreased in the weeks of 30 March and 27 April, but the rank increased. In other cities, both the rank and the proportion of grocery stores visitors increased. We also found that health and personal care stores (NAICS code: 4461) POIs and general merchandise stores, including warehouse clubs and supercentres (NAICS code: 4523) POI showed an upward trend in most of the cities after the outbreak started. For example, the healthcare POI ranked among the top seven in the weeks of 30 March and 27 April in New York and Austin. In Seattle, the proportion of visitors to health and personal care stores POI rose to fifth place in ranking in the weeks of 30 March and 27 April. The proportion of visits to the merchandise POI rose to the top seven in the weeks of 30 March and 27 April in Houston, Dallas, Detroit, Phoenix and rose to the top three in Jacksonville and San Antonio.

Because we determined POI hotspots based on the total number of visitors to POIs, the evolution of the proportion of visitors to POIs in hotspots could provide insights about movement patterns of people across different cities. The results showed that although the absolute number of visitors decreased for all the POIs during COVID-19, the proportion of visitors to restaurants and museums remained dominant in most cities. Also, the results showed that the proportion of visitors to grocery stores and healthcare facilities increased, while the proportion of visitors to amusement and recreation industries decreased. Furthermore, the patterns of visits to POIs did not show a relationship with city categories based on movements to hotspots. Instead, the visits to POIs highly depended on the attributes of cities. For example, gasoline station was the second highest visited POI hotspot in Houston and was third in Dallas and Detroit, while representing only a small proportion of hotspot POI visitors in New York. Museums, historical sites and similar institutions was the second highest proportion of hotspot POI visitors in most studied cities, such as Dallas, Detroit, Philadelphia, Los Angeles and San Jose, while it formed a small proportion of hotspot POI visitors in Jacksonville, Fort Worth and Houston.

## 3.3. Points-of-interest with significant flux-in changes

Based on the number of nodes with significant flux-in changes, we identified several businesses highly affected by the COVID-19 pandemic, including restaurants and other eating places, museums, historical sites and similar institutions, lessors of real estate, elementary and secondary schools (NAICS code: 6111), support activities for air transportation (NAICS code: 4881) and religious organizations (NAICS code: 8131). Also, some of the affected POIs varied across the 16 cities. Figure 7 illustrates the POIs with significant flux-in changes in four selected cities: New York, Austin, San Francisco and Seattle. (The results for other cities can be found in the electronic supplementary material.) We selected one week at the end of each of January, February, March and April to compare the trends with the week of 13 January (with the assumption that most businesses had returned to normal schedules and patterns of visits after the winter break).

Figure 7 illustrates that visitors to restaurants and other eating places and visitors to museums, historical sites, and similar institution POIs were greatly affected in all four cities. These two POIs

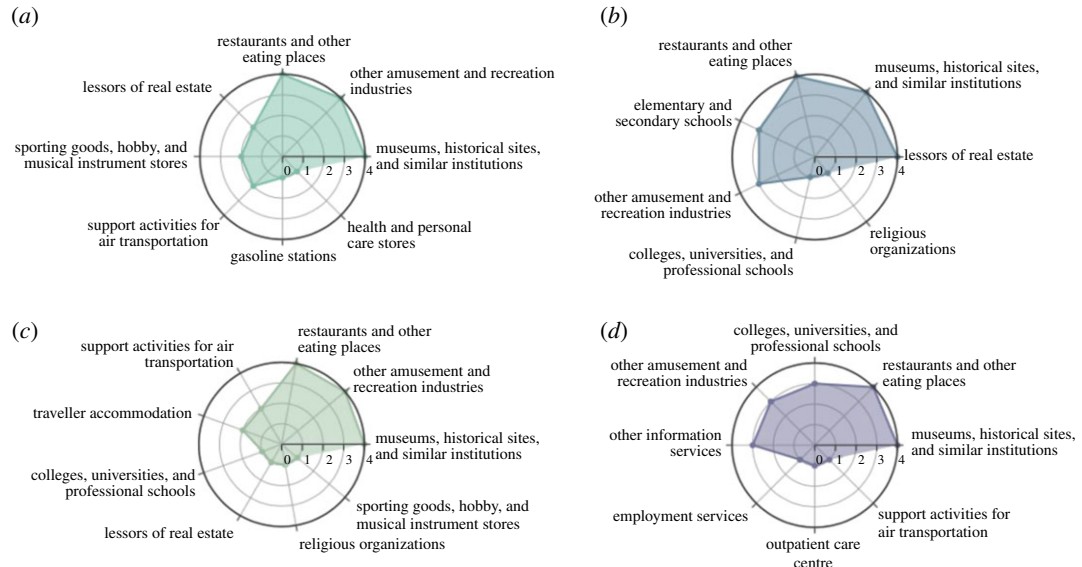

**Figure 7.** POIs with significant flux-in changes in the four one-week periods: (*a*) New York, (*b*) Austin, (*c*) San Francisco and (*d*) Seattle. We included top five businesses related to hotspot POIs with significant flux-in changes in each week. The indicator on the radar chart refers to the number of weeks the business activity was in the top five affected POIs.

ranked in the top five affected business activities across all four studied weeks. Other amusement and recreation industries was another highly affected POI, ranking in the top five affected POIs four times in New York and San Francisco, and in the top five affected POIs list three times in Austin and Seattle. Also, the extent of affected POIs varied across different cities, such as lessors of real estate and elementary and secondary schools in Austin, support activities for air transportation in New York and San Francisco, and college, university and professional schools and other information services in Seattle.

The results indicate that some POIs are universally affected across all cities during the January to May period examined in this study. The effects of the pandemic on other POIs varied across cities and months. For example, the effect on support activities for air transportation visits was related to travel restrictions which had the greatest impact on New York and San Francisco. We can observe from figure 6 that travel accommodation had a relatively large proportion of POI hotspots in New York and San Francisco (ranked top four and five, respectively, before March). Also, the shelter-in-place order affected elementary and secondary schools POIs in Austin and College, University and Professional Schools POIs in Seattle mainly due to closure of schools and colleges.

## 4. Discussion

In this paper, we focused on the patterns of visits to POIs from CBGs during the COVID-19 pandemic. The results of this study provide a deeper insight into the effect of social distancing on changes in population visits to hotspot POIs during the COVID-19 pandemic. The results showed that the absolute number of visitors to POIs showed a downward trend in the 16 studied cities. One category of cities sustained the proportion of movements to hotspot POIs, while another category of cities reduced the proportion of movements to hotspot POIs and increased the proportion of movements to non-hotspots POIs. Another COVID-19 study in Italy demonstrated that human mobility in Italy was strongly related to the spread and control of COVID-19 [32]. Movements to hotspot and non-hotspot, however, may have different transmission risks and cause different epidemic diffusion patterns. Balcan *et al.* [10] considered two types of mobility: long-range mobility and short-range mobility when building an epidemic model. The results showed that two types of mobility determined different epidemic diffusion patterns at regional and local levels. Meloni *et al.* [12] showed that changes of mobility patterns due to an epidemic outbreak may have a negative effect on epidemic control. Furthermore, Chang *et al.* [15] identified 'superspreader' POIs (e.g. fitness centres and restaurants) that may cause a huge amount of infections. Hence, the proportion of movements to urban hotspot POIs could be an important indicator of the manner in which cities respond to an epidemic breakout. This study could contribute to a better theoretical understanding of urban

movement patterns and the effects of mobility reduction policies. The results of the study also facilitate better monitoring of the effect of enforced epidemic control measures. Furthermore, we investigated which POIs maintained their pre-epidemic proportion of visitors, and which POIs experienced declines and increases in the proportion of visitors during the unfolding of COVID-19 and the enforcement of shelter-in-place orders. The results facilitate a better understanding of human lifestyles and their changes during the epidemic, which could help decision-makers to develop effective epidemic control measures.

Also, we conducted chi-square tests to pinpoint POIs with significant flux-in changes. The process could be a good complement to the coarse-grain approach that was adopted to analyse the CBG-POI movement network. The coarse-grain approach clustered nodes to hotspots and non-hotspots and grouped individual CBG-POI flows into four types of movements. While the approach could provide a useful picture of human movements among hotspots and non-hotspots, it cannot provide information about single POIs. The understanding of the flux-in changes for single POIs is important for the examination of pandemics. Because our study focused on the effects of social distancing measures and shelter-in-place orders, the POIs with significant flux-in changes showed decreased visitors during the studied period. This set of results could provide additional insights regarding community response to COVID-19 and help monitor the control measure effectiveness. On the other hand, these POIs could expect significant flux-in increases after the shelter-in-place orders are lifted. Specifying these POIs could provide valuable information to develop reopening policies and strategies (e.g. multi-steps to reopen POIs with significant flux-in changes).

Other research directions could be explored based on the findings of this study. For example, based on the results of the proportions of visitors to POIs during the studied period across cities, we could refine the understanding of essential and non-essential services for humans in urban disruptions, such as natural hazards and epidemic outbreaks [33–35], and future research could take characteristics of cities into consideration. Furthermore, the results could facilitate understanding how the urban disruptions would affect business (e.g. what business industries would be more affected during disruption compared with other business), helping to develop business disaster planning and recovery strategies in urban disruptions [36,37].

The research also has some limitations. The results of the study were based on social distancing and shelter-in-place orders in the United States. In other countries with different policies and cultures, the results may be different. Also, the data cannot consider the interactions among POIs. Decreased visits to one POI may affect visits to another POI. Furthermore, we tried to study movement patterns in some less-populated cities in the United States that were highly affected by COVID-19, such as Randolph, Terrell and Early in Georgia, as well as Union, Bergen and Hudson in New Jersey. The movement data in these cities, however, were very sparse and difficult to build the CBG-POI movement network. The results in this paper, therefore, focused on cities with large populations.

Data accessibility. Data and relevant code for this research work are stored in GitHub: https://github.com/Qingchun-Li/COVID-Movement-Pattern-Analysis and have been archived within the Zenodo Repository: https://doi.org/10.5281/zenodo.4290687.

Authors' contributions. Q.L., L.B. and A.M. were involved in research design and conceptualization; Q.L, L.B., X.X., C.F. and X.G. were involved in data collection, processing, analysis and visualization; Q.L. and A.M. were involved in writing; all authors contributed in reviewing and revising.

Competing interests. We declare we have no competing interests.

Funding. The work was supported by the National Science Foundation RAPID project no. 2026814: Urban Resilience to Health Emergencies: Revealing Latent Epidemic Spread Risks from Population Activity Fluctuations and Collective Sense-making. Any opinions, findings and conclusion or recommendations expressed in this research are those of the authors and do not necessarily reflect the view of the funding agency.

Acknowledgements. The authors would like to acknowledge that SafeGraph provided POI data. The authors would like to thank Jan Gerston for copy-editing services.

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
