## [Reviewer comments · Royal Society Open Science]

Review History

RSOS-201209.R0 (Original submission)

Review form: Reviewer 1

Is the manuscript scientifically sound in its present form?

Yes

Are the interpretations and conclusions justified by the results?

Yes

Is the language acceptable?

No

Do you have any ethical concerns with this paper?

No

Have you any concerns about statistical analyses in this paper?

No

Recommendation?

Major revision is needed (please make suggestions in comments)

Comments to the Author(s)

An interesting paper looking at the impact of COVID-19 on the POI visits.

If I understood correctly, your OD matrix consists of census block groups (CBGs) as origins and POIs as destinations, right? Isn't there is a spatial aggregation disparity here where census blocks represent a different spatial resolution in comparison to POIs. I am not quite sure an OD matrix structured this way provides an accurate picture of mobility in a city.

From Figure 1, I understand that the POIs are not specifically mapped to their locations. Instead the POIs are classified based on a business category. That makes the OD matrix and the resulting coarse grained matrix a bit different from what we usually see in mobility studies. So the origins are still census blocks with mapped geolocation but destinations are now business categories with no geolocation mapping. Am I right? The description of the data and how the OD matrix is constructed is a bit confusing and unclear.

The English quality varies throughout the paper. There are some incorrect or awkward use of English in some locations. Example in Figure 3 caption saying "We used the rolling mean (window = 4) to smooth the data, original data could refer to supplemental document." The second part of this sentence is grammatically wrong and could be re-written in a different and separate way. Please proofread the paper more carefully.

In Figure 3, there is no need to draw the thick red horizontal line. Also, the classification of cities to two groups sounds more like a subjective and qualitative classification. Better make it more systematic.

Figure 3 doesn't have axis titles. Same issue with Figure 4.

The remainder of the paper provides interesting insights into the mobility patterns to different POI categories.

A few suggested references that could strengthen your literature review

<https://doi.org/10.1140/epjds/s13688-017-0129-1>

<https://doi.org/10.1007/s11116-016-9706-6>

<https://doi.org/10.1073/pnas.1203882109>

Overall, the quality of the figures in the main text and supp info could improve. Use a higher resolution image please.

Review form: Reviewer 2

Is the manuscript scientifically sound in its present form?

No

Are the interpretations and conclusions justified by the results?

No

Is the language acceptable?

Yes

Do you have any ethical concerns with this paper?

No

Have you any concerns about statistical analyses in this paper?

Yes

Recommendation?

Major revision is needed (please make suggestions in comments)

Comments to the Author(s)

The manuscript "Disparate Patterns of Movements and Visits to Points of Interest Located in Urban Hotspots across U.S. Metropolitan Cities during COVID-19" by Qingchun Li, Liam Bessell, Xin Xiao, Chao Fan, Xinyu Gao, Ali Mostafavi is a promising contribution to the current efforts of analyzing COVID-19 non-pharmaceutical interventions and their intended and possibly unintended effects. In their paper, the authors examine the movement patterns to urban hotspots in the largest US cities (by population) during the initial 2020 COVID19 outbreak. My comments are listed below. Some of them are rather formal (asking for clarifications) while others are substantial. Hopefully, this evaluation will contribute to the improvement of the manuscript.

#1: This is just a minor suggestion. The title of the paper should indicate the study's design with a commonly used term.

#2: The abstract is balanced and it properly illustrates what was done and what was found. Nevertheless, a brief reference to the limits as well as to the relevance of the study is more than welcome. Authors are also invited to put their work in the international pandemic context.

#3: This is a more substantial comment. Authors are invited to elaborate on the rationale of their study. It is not clear in the introduction section, for instance, the need for their study, the novelty of their study etc. Moreover, what is the contribution of their research? And which are or should be the practical implications of their findings?

#4: Authors need to justify using the modified coarse-grain approach.

#5: In the introduction section, authors refer to "hotspots" and "non-hotspots". It is not clear which were the criteria to differentiate between the two categories. Moreover, it is not clear whether they used a population or a sample of hotspots / non-hotspots.

#6: In Fig 1, (a) and (b) are not very informative. Authors are invited to use other types of visualization.

#7: Authors mention that Fig 1 illustrates the "conceptual model of four types of movements". However, they do not elaborate on the conceptual dimension of the four categories. For instance, what is a "Hotspot CBG"? and how was it theoretically / conceptually created?

#8: Authors refer, on page 4, at "Weekly Pattern Version 2". An in-text clarification of that would ease the reading of the paper.

#9: Why did the authors decide to analyze the top 14 largest US cities by population? Moreover, why did not the authors use another selection criterion?

#10: It is not clear why Detroit was selected. Authors say it "had a burst in the number of cases in March 2020." But they do not explain the country related significance of that.

#11: Fig 2 is not very informative. As in the case of my prior comment (to Fig 1), authors are invited to use another layout so that it will clearly stress what authors want to communicate.

#12: Based on the description provided on page 5, it follows that the "hotspots / non-hotspots" were computationally created and not theoretically derived. If this is the case, authors need to clarify comment #7. In addition, what are the limits of the technical procedure they applied (the identification method) in building the hotspots / non-hotspots? Applying other methods would lead to different results?

- #13: Authors do not justify the selection of the coarse-grain approach to examine the mobility patterns. Also, how much does the approach influence the results / findings?
- #14: Does the population size differences between the cities affect the results? Also, in Fig. 3, why did not the authors normalized the data? Or use the same scale for the comparisons?
- #15: How did the authors address the risk of their results being an artefact of their employed methodology?
- #16: In the Discussion section, the authors do not provide explanations for their research findings. For instance, why some POIs “are universally affected across all cities during the January through May period”? Or why “the proportion of visits to restaurants and museums remained dominant in most cities” etc.
- #17: In what sense the reported results are culturally dependent / case specific?
- #18: Authors mention on page 13 that “Specifying these POIs could provide valuable information to develop reopening policies and strategies (e.g., multi-steps to reopen POIs with significant flux-in changes”. However, they do not explain their findings and in what sense the results can be an input into COVID19 prevention measure relaxation policies.
- #19: Are there any theoretical implications of the reported findings?
- #20: What are the weaknesses of the reported study? Authors do not address this aspect. They only claim that their methodology was not appropriate for less populated cities (page 13).

Decision letter (RSOS-201209.R0)

Dear Mr Li

The Editors assigned to your paper RSOS-201209 "Disparate Patterns of Movements and Visits to Points of Interest Located in Urban Hotspots across U.S. Metropolitan Cities during COVID-19" have now received comments from reviewers and would like you to revise the paper in accordance with the reviewer comments and any comments from the Editors. Please note this decision does not guarantee eventual acceptance.

Please submit your revised manuscript and required files (see below) no later than 21 days from today's (ie 29-Sep-2020) date. Note: the ScholarOne system will 'lock' if submission of the revision is attempted 21 or more days after the deadline. If you do not think you will be able to meet this deadline please contact the editorial office immediately.

Please note article processing charges apply to papers accepted for publication in Royal Society Open Science (<https://royalsocietypublishing.org/rsos/charges>). Charges will also apply to papers transferred to the journal from other Royal Society Publishing journals, as well as papers

submitted as part of our collaboration with the Royal Society of Chemistry (<https://royalsocietypublishing.org/rsos/chemistry>). Fee waivers are available but must be requested when you submit your revision (<https://royalsocietypublishing.org/rsos/waivers>).

Kind regards,
Lianne Parkhouse
Editorial Coordinator
Royal Society Open Science
openscience@royalsociety.org

on behalf of the Associate Editor, and Professor R. Kerry Rowe (Subject Editor)
openscience@royalsociety.org

Associate Editor Comments to Author:

Thank you for your patience while we sought review of this manuscript: given the large (and growing) volume of submissions exploring COVID, it has been difficult to secure the time of very busy researchers to assist. Nevertheless, two reviewers have now commented on your paper. They have identified a number of problems to be addressed, ranging from the analytical to the presentational. Please ensure you thoroughly respond to their concerns and modify your manuscript accordingly, as well as detailing your responses in a point-by-point response document. Furthermore, you are asked to seek advice on the written English - we sympathise that it is not a language known for its logic, but services exist (such as <https://royalsociety.org/journals/authors/benefits/language-editing/>) to assist.

Reviewer comments to Author:

Reviewer: 1

Comments to the Author(s)

An interesting paper looking at the impact of COVID-19 on the POI visits.

If I understood correctly, your OD matrix consists of census block groups (CBGs) as origins and POIs as destinations, right? Isn't there is a spatial aggregation disparity here where census blocks represent a different spatial resolution in comparison to POIs. I am not quite sure an OD matrix structured this way provides an accurate picture of mobility in a city.

From Figure 1, I understand that the POIs are not specifically mapped to their locations. Instead the POIs are classified based on a business category. That makes the OD matrix and the resulting coarse grained matrix a bit different from what we usually see in mobility studies. So the origins are still census blocks with mapped geolocation but destinations are now business categories with no geolocation mapping. Am I right? The description of the data and how the OD matrix is constructed is a bit confusing and unclear.

The English quality varies throughout the paper. There are some incorrect or awkward use of English in some locations. Example in Figure 3 caption saying "We used the rolling mean (window = 4) to smooth the data, original data could refer to supplemental document." The second part of this sentence is grammatically wrong and could be re-written in a different and separate way. Please proofread the paper more carefully.

In Figure 3, there is no need to draw the thick red horizontal line. Also, the classification of cities to two groups sounds more like a subjective and qualitative classification. Better make it more systematic.

Figure 3 doesn't have axis titles. Same issue with Figure 4.

The remainder of the paper provides interesting insights into the mobility patterns to different POI categories.

A few suggested references that could strengthen your literature review

<https://doi.org/10.1140/epjds/s13688-017-0129-1>

<https://doi.org/10.1007/s11116-016-9706-6>

<https://doi.org/10.1073/pnas.1203882109>

Overall, the quality of the figures in the main text and supp info could improve. Use a higher resolution image please.

Reviewer: 2

Comments to the Author(s)

The manuscript “Disparate Patterns of Movements and Visits to Points of Interest Located in Urban Hotspots across U.S. Metropolitan Cities during COVID-19” by Qingchun Li, Liam Bessell, Xin Xiao, Chao Fan, Xinyu Gao, Ali Mostafavi is a promising contribution to the current efforts of analyzing COVID-19 non-pharmaceutical interventions and their intended and possibly unintended effects. In their paper, the authors examine the movement patterns to urban hotspots in the largest US cities (by population) during the initial 2020 COVID19 outbreak. My comments are listed below. Some of them are rather formal (asking for clarifications) while others are substantial. Hopefully, this evaluation will contribute to the improvement of the manuscript.

#1: This is just a minor suggestion. The title of the paper should indicate the study’s design with a commonly used term.

#2: The abstract is balanced and it properly illustrates what was done and what was found. Nevertheless, a brief reference to the limits as well as to the relevance of the study is more than welcome. Authors are also invited to put their work in the international pandemic context.

#3: This is a more substantial comment. Authors are invited to elaborate on the rationale of their study. It is not clear in the introduction section, for instance, the need for their study, the novelty of their study etc. Moreover, what is the contribution of their research? And which are or should be the practical implications of their findings?

#4: Authors need to justify using the modified coarse-grain approach.

#5: In the introduction section, authors refer to “hotspots” and “non-hotspots”. It is not clear which were the criteria to differentiate between the two categories. Moreover, it is not clear whether they used a population or a sample of hotspots / non-hotspots.

#6: In Fig 1, (a) and (b) are not very informative. Authors are invited to use other types of visualization.

#7: Authors mention that Fig 1 illustrates the “conceptual model of four types of movements”. However, they do not elaborate on the conceptual dimension of the four categories. For instance, what is a “Hotspot CBG”? and how was it theoretically / conceptually created?

#8: Authors refer, on page 4, at “Weekly Pattern Version 2”. An in-text clarification of that would ease the reading of the paper.

#9: Why did the authors decide to analyze the top 14 largest US cities by population? Moreover, why did not the authors use another selection criterion?

#10: It is not clear why Detroit was selected. Authors say it “had a burst in the number of cases in March 2020.” But they do not explain the country related significance of that.

#11: Fig 2 is not very informative. As in the case of my prior comment (to Fig 1), authors are invited to use another layout so that it will clearly stress what authors want to communicate.

#12: Based on the description provided on page 5, it follows that the “hotspots / non-hotspots” were computationally created and not theoretically derived. If this is the case, authors need to clarify comment #7. In addition, what are the limits of the technical procedure they applied (the identification method) in building the hotspots /non-hotspots? Applying other methods would conduct to different results?

#13: Authors do not justify the selection of the coarse-grain approach to examine the mobility patterns. Also, how much does the approach influence the results / findings?

#14: Does the population size differences between the cities affect the results? Also, in Fig. 3, why did not the authors normalized the data? Or use the same scale for the comparisons?

#15: How did the authors address the risk of their results being an artefact of their employed methodology?

#16: In the Discussion section, the authors do not provide explanations for their research findings. For instance, why some POIs “are universally affected across all cities during the January through May period”? Or why “the proportion of visits to restaurants and museums remained dominant in most cities” etc.

#17: In what sense the reported results are culturally dependent / case specific?

#18: Authors mention on page 13 that “Specifying these POIs could provide valuable information to develop reopening policies and strategies (e.g., multi-steps to reopen POIs with significant flux-in changes”. However, they do not explain their findings and in what sense the results can be an input into COVID19 prevention measure relaxation policies.

#19: Are there any theoretical implications of the reported findings?

#20: What are the weaknesses of the reported study? Authors do not address this aspect. They only claim that their methodology was not appropriate for less populated cities (page 13).

===PREPARING YOUR MANUSCRIPT===

If you have been asked to revise the written English in your submission as a condition of publication, you must do so, and you are expected to provide evidence that you have received

language editing support. The journal would prefer that you use a professional language editing service and provide a certificate of editing, but a signed letter from a colleague who is a native speaker of English is acceptable. Note the journal has arranged a number of discounts for authors using professional language editing services (<https://royalsociety.org/journals/authors/benefits/language-editing/>).

===PREPARING YOUR REVISION IN SCHOLARONE===

<https://royalsociety.org/journals/authors/author-guidelines/#supplementary-material> to

include a suitable title and informative caption. An example of appropriate titling and captioning may be found at https://figshare.com/articles/Table_S2_from_Is_there_a_trade-off_between_peak_performance_and_performance_breadth_across_temperatures_for_aerobic_sc_ope_in_teleost_fishes_/3843624.

Author's Response to Decision Letter for (RSOS-201209.R0)

See Appendix A.

RSOS-201209.R1 (Revision)

Review form: Reviewer 1

Is the manuscript scientifically sound in its present form?

Yes

Are the interpretations and conclusions justified by the results?

Yes

Is the language acceptable?

Yes

Do you have any ethical concerns with this paper?

No

Have you any concerns about statistical analyses in this paper?

No

Recommendation?

Accept as is

Comments to the Author(s)

Authors have addressed the reviewers' comments to a satisfactory level.

Review form: Reviewer 2

Is the manuscript scientifically sound in its present form?

Yes

Are the interpretations and conclusions justified by the results?

Yes

Is the language acceptable?

Yes

Do you have any ethical concerns with this paper?

No

Have you any concerns about statistical analyses in this paper?

No

Recommendation?

Accept as is

Comments to the Author(s)

On my view, all the points raised during the previous round of review were addressed properly and in a rigorous manner. On my view, the manuscript may be considered for publication.

Decision letter (RSOS-201209.R1)

Dear Mr Li,

It is a pleasure to accept your manuscript entitled "Disparate Patterns of Movements and Visits to Points of Interest Located in Urban Hotspots across U.S. Metropolitan Cities during COVID-19" in its current form for publication in Royal Society Open Science. The comments of the reviewer(s) who reviewed your manuscript are included at the foot of this letter.

At this stage, we ask that you please archive your GitHub code within the Zenodo repository: <https://guides.github.com/activities/citable-code/>. By doing this, a formal, citable DOI will be associated with your data record, and an open license (CC-BY preferred) can be applied to your data. We would then ask that you please update your data availability statement to read as:

"Data and relevant code for this research work are stored in GitHub: [GitHub URL here] and have been archived within the Zenodo repository: <https://doi.org/zenodo.....> [ref number].

COVID-19 rapid publication process:

We are taking steps to expedite the publication of research relevant to the pandemic. If you wish, you can opt to have your paper published as soon as it is ready, rather than waiting for it to be published the scheduled Wednesday.

This means your paper will not be included in the weekly media round-up which the Society sends to journalists ahead of publication. However, it will still appear in the COVID-19 Publishing Collection which journalists will be directed to each week (<https://royalsocietypublishing.org/topic/special-collections/novel-coronavirus-outbreak>).

If you wish to have your paper considered for immediate publication, or to discuss further, please notify openscience_proofs@royalsociety.org and press@royalsociety.org when you respond to this email.

on behalf of Prof R. Kerry Rowe (Subject Editor)
openscience@royalsociety.org

Reviewer comments to Author:
Reviewer: 1

Comments to the Author(s)
Authors have addressed the reviewers' comments to a satisfactory level.

Reviewer: 2

Comments to the Author(s)
On my view, all the points raised during the previous round of review were addressed properly and in a rigorous manner. On my view, the manuscript may be considered for publication.

Appendix A

Comments of Reviewers

Reviewer #1:

I. An interesting paper looking at the impact of COVID-19 on the POI visits. If I understood correctly, your OD matrix consists of census block groups (CBGs) as origins and POIs as destinations, right? Isn't there is a spatial aggregation disparity here where census blocks represent a different spatial resolution in comparison to POIs. I am not quite sure an OD matrix structured this way provides an accurate picture of mobility in a city.

Authors' Response:

The authors appreciate reviewer's insightful comments. The reviewer is right that the OD matrix consists of CBGs as origins and POIs as destinations. Essentially, the OD matrix in this paper is the CBG-POI matrix. The authors concur with reviewer's comments that the spatial resolution of origins and destinations are different, and the CBG-POI matrix includes heterogeneous types of nodes. Due to the limitation of SafeGraph POI data, the structured CBG-POI matrix can only reflect the mobility from CBGs to POIs. In this paper, therefore, we focused on the pattern of visits to POIs (including POI hotspots and non-hotspots) instead of the total mobility in a city. Also, the method employed in the paper calculated the hotspots and non-hotspots of origins and destinations separately. Hence, the different spatial resolutions of origins and destinations would not affect the determination of hotspots and non-hotspots in origins and destinations. Therefore, we believe that the structured CBG-POI matrix could reflect the pattern of visits to POIs from CBGs, which was the main objective of this analysis. In the revised manuscript, according to the reviewer's comments, we elaborated on the CBG-POI movement network representation and clarified that the paper focused on the pattern of visits to POIs from CBGs and the employed method is insensitive to the different spatial resolution of CBGs and POIs.

Manuscript Updates:

In Methodology part:

“In this study, we focused on the patterns of visits to POIs from CBGs to examine movement patterns across hotspot and non-hotspot clusters. Therefore, the coarse-grain approach that clusters hotspot and non-hotspot nodes in CBG-POI movement networks were used [1,14,15]. In the existing literature, different methods have been proposed to separate hotspots and non-hotspots. Louail et al. developed a method to separate hotspots and non-hotspots based on the Lorenz curve of divided 1-km^2 cells [14,15]. This method yields lower and upper boundaries of hotspots. Hamedmoghadam et al. showed that using Lorenz curve to identify hotspots and non-hotspots was biased to the outlier nodes [1]. Therefore, they proposed a modified coarse-grain approach using a centroid-based clustering method to separate hotspots and non-hotspots. In this paper, we adopted the modified coarse-grain approach that separated hotspot and non-hotspot nodes in the mapped CBG-POI movement networks. The employed method determined hotspot and non-hotspot nodes in CBGs and POIs separately. Therefore, different spatial resolutions of CBGs and POIs will not affect the results of hotspot and non-hotspot nodes in CBGs and POIs.”

Page (32), line (41-55)

In Discussion part:

“In this paper, we focused on the patterns of visits to POIs from CBGs during the COVID-19 pandemic. The results of this study could provide a deeper insight into the effects of social distancing on changes in visits to hotspot POIs from CBGs during the COVID-19 pandemic.”

Page (41), line (3-6)

2. From Figure 1, I understand that the POIs are not specifically mapped to their locations. Instead the POIs are classified based on a business category. That makes the OD matrix and the resulting coarse grained matrix a bit different from what we usually see in mobility studies. So the origins are still census blocks with mapped geolocation but destinations are now business categories with no geolocation mapping. Am I right? The description of the data and how the OD matrix is constructed is a bit confusing and unclear.

Authors' Response:

Thank for the reviewer's comments. We clarified the data and method in the revised manuscript. In Figure 1, the POIs are mapped to their locations. For the CBG-POI matrix, the origins are CBGs with mapped geolocation and destinations are POIs with mapped geolocation as well. Based on this, we used the coarse-grained method to study the patterns of four types of visits to POIs from CBGs (i.e., hotspots to hotspots, non-hotspots to hotspots, hotspots to non-hotspots and non-hotspots to non-hotspots). Furthermore, because SafeGraph provides the business category of each POI, we also investigated proportions of persons visiting to different business categories based on the visits to POIs from CBGs. We revised the description of data and how the CBG-POI matrix is constructed in the revised manuscript based on reviewer's comments. Also, we revised the caption of Figure 1 to indicate that the nodes are visualized based on their geolocations.

Manuscript Updates:

Caption for Figure 1:

“The coarse-grain approach categorizes origin-destination movements to four types of movements among hotspots and non-hotspots: (a) hotspots (red nodes) and non-hotspots (blue nodes); nodes are visualized based on their geolocations, (b) individual CBG-POI movements among hotspots and non-hotspots, four colors represent four types of movements, (c) clarification of four types of movements among hotspots and non-hotspots, (Figure 1 was plotted based on the SafeGraph data for Houston.)”

Page (31), line (14-20),

In the “Data and Methodology” section:

“We mapped the CBG-POI movement networks based on the number of visitors to POIs from CBGs. The mapped CBG-POI movement networks are directed and weighted bipartite networks, where pairs of nodes i and j represent CBGs and POIs with mapped geolocation. Links in the CBG-POI movement networks represent visits from CBGs to POIs, and non-negative weights of links $w_{ij} > 0$ represent one or more visitors during the covered period. If there was no movement from CBGs to POIs, $w_{ij} = 0$.”

Page (32), line (3-13)

3. *The English quality varies throughout the paper. There are some incorrect or awkward use of English in some locations. Example in Figure 3 caption saying "We used the rolling mean (window = 4) to smooth the data, original data could refer to supplemental document." The second part of this sentence is grammatically wrong and could be re-written in a different and separate way. Please proofread the paper more carefully.*

Authors' Response:

The authors thank for reviewer's comments. We adjusted the second part of the sentence that the reviewer mentioned, and carefully proofread the paper again.

Manuscript Updates:

Figure 3 caption: "We used the rolling mean (window = 4) to smooth the data. For the original data, please see the supplemental document."

Page (36), line (32-33)

4. *In Figure 3, there is no need to draw the thick red horizontal line. Also, the classification of cities to two groups sounds more like a subjective and qualitative classification. Better make it more systematic.*

Authors' Response:

The authors are thankful for the reviewer's comments. We removed the thick red horizontal line in the revised manuscript according to the reviewer's comments.

The classification of cities into two groups was according to the clustering algorithm based on the time series of visits to POIs. "We conducted the clustering analysis after we obtained four types of movements among hotspots and non-hotspots in 16 cities. We compared three algorithms (Euclidean distances, dynamic time warping (DTW), cross correlation) for time series clustering [2–4] and used the silhouette coefficient to determine the number of clusters [5]. (Results of the algorithms are presented in the supplementary material.)" Please see the original manuscript, page 6 (page number generated by the submission system instead of the page number in the manuscript), line 42-51.

Manuscript Updates:

Figure 3 and Figure 4

Page (36, 37)

5. *Figure 3 doesn't have axis titles. Same issue with Figure 4..*

Authors' Response:

Thanks for the reviewer's comments. We added axis titles for Figure 3 and Figure 4 in the revised manuscript. Because Figure 3 and Figure 4 have 16 subplots that shared the same axis titles, we added axis titles for the whole figures.

Manuscript Updates:

Figure 3 and Figure 4

Page (36, 37)

6. The remainder of the paper provides interesting insights into the mobility patterns to different POI categories.

A few suggested references that could strengthen your literature review

<https://doi.org/10.1140/epjds/s13688-017-0129-1>

<https://doi.org/10.1007/s11116-016-9706-6>

<https://doi.org/10.1073/pnas.1203882109>

Authors' Response:

The authors thank reviewer for the good suggestions. We read through the references and discussed/cited them in the introduction and discussion parts in the revised manuscript.

Manuscript Updates:

“Urban mobility and movement patterns are important characteristics of urban dynamics, reflecting the collective human behavior and social interactions [1,6].”

Page (29), line (8-10),

“Understanding mobility patterns of the visits to urban hotspots is important for developing and monitoring epidemic/pandemic control measures. Origin-destination (OD) network analysis provides a powerful tool to study mobility patterns under such a situation, and are especially helpful for locating hotspots and studying the urban mobility patterns of visiting urban hotspots [7,8].”

Page (29), line (47-52),

“Urban mobility drives the spatial flux of populations, and effective disaster and epidemic response would greatly benefit from the characterization of urban mobility patterns [9–13].”

Page (29), line (10-12)

7. Overall, the quality of the figures in the main text and supp info could improve. Use a higher resolution image please.

Authors' Response:

Thank for the helpful comments. It seems that the submission system degraded the quality of the figures in the manuscript. In this round, we submitted all the high-resolution Figures to the system separately and the journal could use the high resolution images if needed.

Reviewer #2:

1. This is just a minor suggestion. The title of the paper should indicate the study's design with a commonly used term.

Authors' Response:

The authors appreciate the reviewer's suggestion. In the original title, we try to note the results of the study.

2. The abstract is balanced, and it properly illustrates what was done and what was found. Nevertheless, a brief reference to the limits as well as to the relevance of the study is more than welcome. Authors are also invited to put their work the international pandemic context.

Authors' Response:

Thank you. We modified the abstract according to reviewer's suggestions.

Manuscript Updates:

"The study was limited to 16 metropolitan cities in the United States. The proposed methodology could be applied to digital trace data in other cities and countries to study the patterns of movements to POIs during the COVID-19 pandemic."

Page (28), line (49-52),

3. This is a more substantial comment. Authors are invited to elaborate on the rationale of their study. It is not clear in the introduction section, for instance, the need for their study, the novelty of their study etc. Moreover, what is the contribution of their research? And which are or should be the practical implications of their findings?

Authors' Response:

The authors appreciate the reviewer's constructive comments. According to the reviewer's comments, we added statements to elaborate the rationale of the study in the introduction part in the revised manuscript.

Manuscript Updates:

"In summary, the extant studies demonstrated that the characterization of urban mobility and movement patterns are important to understand the collective human behavior and social interactions, which are critical for the development of effective epidemic control measures. Also, urban hotspots are gravity activity centers that usually have higher density of populations and POIs, which could be potential 'superspreaders' during the pandemic situation. Therefore, understanding the patterns of visits to POIs in urban hotspots is important for developing and monitoring epidemic control measures. Extant studies, however, rarely studied the patterns of visits to POIs in urban hotspots under pandemic situations. Hence, in this paper, we investigated the patterns of population visits to urban hotspots using origin-destination networks from census block groups (CBGs) to points of interest (POIs) in 16 cities of United States based on the digital

trace data from SafeGraph. The POI data enable the identification of urban hotspots to evaluate changes in visiting urban hotspots due to social distancing measures during the COVID-19 pandemic. We also identified POIs that had significant flux-in decreases and what POI associated business categories were greatly affected during COVID-19. These POIs and business categories could expect a significant flux-in increase after the shelter-in-place orders are lifted. The results of this study could help decision-makers better monitor and evaluate epidemic/pandemic control measures, as well as reopening policies and strategies.”

Page (30), line (11-23),

4. Authors need to justify using the modified coarse-grain approach.

Authors’ Response:

We added the justification for the modified coarse-grain approach in the revised manuscript according to the reviewer’s comments. We included this detailed explanation in the methodology part.

Manuscript Updates:

“In this study, we focused on the patterns of visits to POIs from CBGs to examine movement patterns across hotspot and non-hotspot clusters. Therefore, the coarse-grain approach that clusters hotspot and non-hotspot nodes in CBG-POI movement networks were used [1,14,15]. In the existing literature, different methods have been proposed to separate hotspots and non-hotspots. Louail et al. [14,15] developed a method to separate hotspots and non-hotspots based on the Lorenz curve of divided 1-km^2 cells. This method yields lower and upper boundaries of hotspots. Hamedmoghadam et al. showed that using Lorenz curve to identify hotspots and non-hotspots was biased to the outlier nodes [1]. Therefore, they proposed a modified coarse-grain approach using a centroid-based clustering method to separate hotspots and non-hotspots. In this paper, we adopted the modified coarse-grain approach that separated hotspot and non-hotspot nodes in the mapped CBG-POI movement networks.”

Page (32), line (41-52)

5. In the introduction section, authors refer to “hotspots” and “non-hotspots”. It is not clear which were the criteria to differentiate between the two categories. Moreover, it is not clear whether they used a population or a sample of hotspots / non-hotspots.

Authors’ Response:

Thank you. We introduced the definitions of “hotspots” and “non-hotspots” in the introduction part and detailly explained how we differentiated the “hotspots” and “non-hotspots” in the methodology part. In the revised manuscript, we also added description in the introduction part to help readers refer to the approach that separates “hotspots” and “non-hotspots” in the methodology part.

Manuscript Updates:

“In this paper, we adopted a modified coarse-grain approach for separating the hotspots and non-hotspots nodes in mapped CBG-POI movement networks [1]. Hotspot and non-hotspot nodes are computationally determined. Hotspots are nodes with higher weighted degree centrality, and non-hotspots are nodes with lower weighted degree centrality in the mapped CBG-POI networks. The adopted method determined the threshold to separate the hotspot and non-hotspot nodes. A detailed explanation of the adopted approach is presented in the methodology part.”

Page (30), line (33-40)

6. *In Fig 1, (a) and (b) are not very informative. Authors are invited to use other types of visualization.*

Authors' Response:

Thank for the reviewer's comments. We used Figure 1 to provide an example of hotspots and non-hotspots in the city and four types of movements among them. In the revised manuscript, we modified the caption of Figure 1 to provide more information and help readers to understand.

Manuscript Updates:

“**Figure 1.** The coarse-grain approach categorizes CBG-POI movements to four types of movements among hotspots and non-hotspots: (a) separated hotspots (red nodes) and non-hotspots (blue nodes), (b) individual CBG-POI movements among hotspots and non-hotspots, four colors represent four types of movements, (c) clarification of four types of movements among hotspots and non-hotspots, (Figure 1 was plotted based on the SafeGraph data for Houston.)”

Page (31), line (14-20)

7. *Authors mention that Fig 1 illustrates the “conceptual model of four types of movements”. However, they do not elaborate on the conceptual dimension of the four categories. For instance, what is a “Hotspot CBG”? and how was it theoretically / conceptually created?*

Authors' Response:

Thank you. As we responded in comment 5 and 6, hotspots and non-hotspots were computationally determined. We rewrote the sentence to remove the confusion and clarify how we determined the hotspots and non-hotspots.

Manuscript Updates:

“In this paper, we adopted a modified coarse-grain approach for separating the hotspots and non-hotspots nodes in mapped CBG-POI movement networks [1]. Hotspot and non-hotspot nodes are computationally determined. Hotspots are nodes with higher weighted degree centrality, and non-hotspots are nodes with lower weighted degree centrality in the mapped CBG-POI networks. The adopted method determines the threshold to separate the hotspot nodes and non-hotspot nodes. A detailed explanation of the adopted approach is presented in the methodology part. Figure 1 illustrates four types of movements among hotspots and non-hotspots.”

Page (30), line (33-44)

8. *Authors refer, on page 4, at “Weekly Pattern Version 2”. An in-text clarification of that would ease the reading of the paper.*

Authors’ Response:

Thanks. We added an in-text clarification of that data in the revised manuscript.

Manuscript Updates:

“We used the POI data: Weekly Pattern Version 2, to study movement patterns in sixteen cities in the United States. The Safegraph weekly pattern data provide information related to the visits to POIs and cover the entire United States. The data were aggregated weekly (Monday to Sunday [16]). “

Page (31), line (35-37)

9. *Why did the authors decide to analyze the top 14 largest US cities by population? Moreover, why did not the authors used another selection criterion?*

Authors’ Response:

Thank you. We had several considerations when decided to analyze the top 14 largest US cities by population: (1) we found that Safegraph collected more data in larger cities. We tested several less populated cities in United States such as Randolph, Terrell, and Early in Georgia, as well as Union, Bergen, and Hudson in New Jersey. The visits to POI data in these cities, however, were sparse and it was difficult to build the OD networks, and (2) we were more concerned about the spread of COVID-19 in cities with larger population. We elaborated these considerations in the revised manuscript.

Manuscript Updates:

“We selected the top 14 largest cities in the United States due to two considerations: (1) Safegraph collected more data in larger cities. We tested several less populated cities in United States such as Randolph, Terrell, and Early in Georgia, as well as Union, Bergen, and Hudson in New Jersey. The visits to POI data in these cities, however, were sparse and it was infeasible to build the CBG-POI movement networks, and (2) we were more concerned about the spread of COVID-19 in cities with larger population.”

Page (31), line (38-44)

10. *It is not clear why Detroit was selected. Authors say it “had a burst in the number of cases in March 2020.” But they do not explain the country related significance of that.*

Authors' Response:

Thanks for the comment. We decided to include Detroit because when we started our study in March 2020, Detroit had a burst in the number of confirmed cases. Also, Detroit is a good study area because it is the largest city in the midwestern state of Michigan. With the study progressing, the spread of COVID-19 in Detroit was effectively controlled and the number of confirmed cases in Detroit greatly decreased. Therefore, Detroit became a good example to compare with other cities that kept the larger number of confirmed cases. We added these reasons in the revised manuscript according to the reviewer's comments.

Manuscript Updates:

"In addition, Seattle and Detroit were studied. Seattle was the first city in the United States to report a diagnosed COVID case, and Detroit had a burst in the number of confirmed cases in March 2020. Also, we considered that Detroit is the largest city in the midwestern state of Michigan, and the number of confirmed cases in Detroit was greatly decreased after April 2020. Therefore, Detroit became a good example to compare with other cities."

Page (31), line (46-49)

11. Fig 2 is not very informative. As in the case of my prior comment (to Fig 1), authors are invited to use another layout so that it will clearly stress what authors want to communicate.

Authors' Response:

Thanks. We tried to use Fig 2 to provide an example of the mapped CBG-POI movement networks. The information that we tried to provide includes how many nodes, including hotspots and non-hotspots, and how many weighted edges. We included the information in the caption of Fig 2.

Manuscript Updates:

"Figure 2. Mapped CBG-POI movement network for the week of January 27, 2020, in Jacksonville, Florida. The figure shows a total of 83,661 weighted edges. Red nodes represent hotspots (1,314 nodes) and blue nodes represent non-hotspots (10,820 nodes)."

Page (32), line (32-35)

12. Based on the description provided on page 5, it follows that the "hotspots / non-hotspots" were computationally created and not theoretically derived. If this is the case, authors need to clarify comment #7. In addition, what are the limits of the technical procedure they applied (the identification method) in building the hotspots /non-hotspots? Applying other methods would conduct to different results?

Authors' Response:

Thanks for the reviewer's comments. We clarified comment #7.

As responded in comment #4, we compared different approaches that separates hotspots and non-hotspots in the literature [1,14,15]. We finally adopted the approach proposed by

Hamedmoghadam et al., because they demonstrated that using Lorenz curve to identify hotspots and non-hotspots was biased to the outlier nodes [1]. Therefore, we believe that the adopted approach is more reasonable than other methods.

Manuscript Updates:

Please refer to the response to comment #4 for the justification of the selection of the modified coarse-grain approach.

13. Authors do not justify the selection of the coarse-grain approach to examine the mobility patterns. Also, how much does the approach influence the results / findings?

Authors' Response:

The authors thank for the reviewer's comments. We justified the selection of the modified coarse-grain approach in the response to comment #4. Also, as responded in comment #12, we believe the adopted approach is more reasonable compared with other methods.

Manuscript Updates:

Please refer to the response to comment #4 and comment #12.

14. Does the population size differences between the cities affect the results? Also, in Fig. 3, why did not the authors normalize the data? Or use the same scale for the comparisons?

Authors' Response:

Thank for the comments. The population size differences between the cities would not affect the results, because we only compare the increasing or decreasing trends instead of absolute numbers among cities. In Fig. 3, normalizing the data by each city's population would not change the trends in each city. Furthermore, in Fig. 3, we showed the absolute numbers of four types of movements in cities, while in Fig. 4, we showed the proportions of four types of movements in all cities. Data shown in Fig. 4 were normalized by the total number of four types of movements in each city. We added following contents in the revised manuscript according to the reviewer's comments.

Manuscript Updates:

“Figure 3 illustrates that the sum of visitors to POIs showed a decreasing trend for all 16 cities after the enforcement of shelter-in-place orders. However, four types of movements (HH, HN, HH, and NH) varied across different cities. Because we only compared the increasing or decreasing trends of four types of movements among cities, the population size differences between the cities would not affect the results. Figure 4 shows the proportion of each type of movements in 16 cities. Data shown in Figure 4 were normalized by the total number of four types of movements in each city.”

Page (34), line (46-55)

15. How did the authors address the risk of their results being an artefact of their employed methodology?

Authors' Response:

The authors thank for the reviewer for this comment. When we designed the study, we addressed the risk of the results being an artefact through following steps.

1. We conducted our study informed by the existing literature. We conducted a comprehensive literature review to compare different methods and finally determine the appropriate methodology. Therefore, we believe that our methodology is theoretically reliable.

2. We used an interdependent, reliable and replicable dataset. Safegraph is an interdependent company that provides POI data. Also, Safegraph provides documents that elaborate on how they collected and dealt with the data. Furthermore, Safegraph provides free access to the POI data for educational institutions and different kinds of research have been conducted based on the POI data provided by Safegraph. Therefore, we believe that the POI data is reliable, transparent and replicable.

16. In the Discussion section, the authors do not provide explanations for their research findings. For instance, why some POIs “are universally affected across all cities during the January through May period”? Or why “the proportion of visits to restaurants and museums remained dominant in most cities” etc.

Authors' Response:

Thank you. We did not provide explanations for these two findings because investigating the real reason for these two results is beyond the scope of this study. However, in the original manuscript, we mentioned that there are future research directions could be explored based on the findings of this study.

“Furthermore, we investigated which POIs maintained their pre-epidemic proportion of visits, and which POIs experienced declines and increases in proportion visits during the unfolding of COVID-19 and the enforcement of shelter-in-place orders. The results facilitate a better understanding of human lifestyles and its changes during the epidemic and could inform developing effective epidemic control measures.” Page 13, line 22-26, original manuscript.

“Other research directions could be explored based on the findings of this study. For example, based on the results of the proportions of visits to POIs during the studied period across cities, we could refine the understanding of essential and non-essential services for humans in urban disruptions, such as natural hazards and epidemic outbreaks [19–21], and future research could take characteristics of cities into consideration.” Page 13, line 42-47, original manuscript.

17. In what sense the reported results are culturally dependent / case specific?

Authors' Response:

Thanks. We agree that the results are based on the social distancing and shelter-in-place orders in the United States. In other countries with different policies, the results may be different. The following contents have been added to the manuscript (the limitation part) according to the reviewer's comments.

Manuscript Updates:

“The results of the study were based on the social distancing and shelter-in-place orders in the United States. In other counties with different policies and cultures, the results may be different.”

Page (42), line (5-8)

18. Authors mention on page 13 that “Specifying these POIs could provide valuable information to develop reopening policies and strategies (e.g., multi-steps to reopen POIs with significant flux-in changes)”. However, they do not explain their findings and in what sense the results can be an input into COVID19 prevention measure relaxation policies.

Authors' Response:

Thank for the comments. In the original manuscript, we explained that the POIs with significant flux-in decreases could expect significant flux-in increases after the shelter-in-place orders are lifted. Therefore, specifying these POIs could provide valuable information to develop reopening policies and strategies (e.g., multi-steps to reopen POIs with significant flux-in changes). Page 13, line 39-40, original manuscript.

19. Are there any theoretical implications of the reported findings?

Authors' Response:

Thank for the comment. We added theoretical implication of the reported findings in the revised manuscript.

Manuscript Updates:

“The proportion of movements to urban hotspot POIs could be an important indicator of the manner in which cities respond to an epidemic breakout. This study could contribute to better theoretical understanding of urban movement patterns from CBGs to POIs and the effects of mobility reduction policies.”

Page (41), line (21-25)

20. What are the weaknesses of the reported study? Authors do not address this aspect. They only claim that their methodology was not appropriate for less populated cities (page 13).

Authors' Response:

Thank you. We discussed more weakness of the study in the revised manuscript.

Manuscript Updates:

“The research also has some limitations. The results of the study were based on the social distancing and shelter-in-place orders in the United States. In other counties with different policies and cultures, the results may be different. Also, the data cannot consider the interactions among POIs. Decreased visits to one POI may affect visits to another POI.”

Page (42), line (4-8)

References in the response (sequence may be different in the manuscript):

1. Hamedmoghadam H, Ramezani M, Saberi M. Revealing latent characteristics of mobility networks with coarse-graining. *Sci Rep.* 2019;9(1).
2. Petitjean F, Ketterlin A, Gançarski P. A global averaging method for dynamic time warping, with applications to clustering. *Pattern Recognit.* 2011;44(3):678–93.
3. Cuturi M, Blondel M. Soft-DTW: A differentiable loss function for time-series. In: 34th International Conference on Machine Learning, ICML 2017. 2017. p. 1483–505.
4. Paparrizos J, Gravano L. K-shape: Efficient and accurate clustering of time series. In: Proceedings of the ACM SIGMOD International Conference on Management of Data. 2015. p. 1855–70.
5. Rousseeuw PJ. Silhouettes: A graphical aid to the interpretation and validation of cluster analysis. *J Comput Appl Math.* 1987;20(C):53–65.
6. Cuttone A, Lehmann S, González MC. Understanding predictability and exploration in human mobility. *EPJ Data Sci.* 2018;7(1).
7. Oliver N, Letouzé E, Sterly H, Delataille S, Nadai M De, Lepri B, et al. Mobile phone data and COVID-19: Missing an opportunity? *arXiv e-prints.* 2020;arXiv:2003.12347 (2020).
8. Saberi M, Mahmassani HS, Brockmann D, Hosseini A. A complex network perspective for characterizing urban travel demand patterns: graph theoretical analysis of large-scale origin–destination demand networks. *Transportation (Amst).* 2017;44(6):1383–402.
9. Danon L, House T, Keeling MJ. The role of routine versus random movements on the spread of disease in Great Britain. *Epidemics.* 2009;1(4):250–8.
10. Le Menach A, Tatem AJ, Cohen JM, Hay SI, Randell H, Patil AP, et al. Travel risk, malaria importation and malaria transmission in Zanzibar. *Sci Rep.* 2011;1.
11. Merler S, Ajelli M. The role of population heterogeneity and human mobility in the spread of pandemic influenza. *Proc R Soc B Biol Sci.* 2010;277(1681):557–65.
12. Wesolowski A, Eagle N, Tatem AJ, Smith DL, Noor AM, Snow RW, et al. Quantifying the impact of human mobility on malaria. *Science (80-).* 2012;338(6104):267–70.

13. Lu X, Bengtsson L, Holme P. Predictability of population displacement after the 2010 Haiti earthquake. *Proc Natl Acad Sci U S A*. 2012;109(29):11576–81.
14. Louail T, Lenormand M, Picornell M, Cantú OG, Herranz R, Frias-Martinez E, et al. Uncovering the spatial structure of mobility networks. *Nat Commun*. 2015;6.
15. Louail T, Lenormand M, Cantu Ros OG, Picornell M, Herranz R, Frias-Martinez E, et al. From mobile phone data to the spatial structure of cities. *Sci Rep*. 2014;4.
16. SafeGraph. Weekly Pattern Version 2 [Internet]. SafeGraph. 2020. Available from: <https://docs.safegraph.com/docs/weekly-patterns>
17. Benzell SG, Collis A, Nicolaides C. Rationing social contact during the COVID-19 pandemic: Transmission risk and social benefits of US locations. *Proc Natl Acad Sci*. 2020;202008025.
18. Chang SY, Pierson E, Koh PW, Gerardin J, Redbird B, Grusky D, et al. Mobility network modeling explains higher SARS-CoV-2 infection rates among disadvantaged groups and informs reopening strategies. *medRxiv*. 2020;2020.06.15.20131979.
19. Esmalian A, Ramaswamy M, Rasoulkhani K, Mostafavi A. Agent-Based Modeling Framework for Simulation of Societal Impacts of Infrastructure Service Disruptions during Disasters. In: *Computing in Civil Engineering 2019: Smart Cities, Sustainability, and Resilience - Selected Papers from the ASCE International Conference on Computing in Civil Engineering 2019*. 2019. p. 16–23.
20. Dargin JS, Mostafavi A. Human-centric infrastructure resilience: Uncovering well-being risk disparity due to infrastructure disruptions in disasters. Linkov I, editor. *PLoS One* [Internet]. 2020 Jun 18;15(6):e0234381. Available from: <https://dx.plos.org/10.1371/journal.pone.0234381>
21. Coleman N, Esmalian A, Mostafavi A. Equitable Resilience in Infrastructure Systems: Empirical Assessment of Disparities in Hardship Experiences of Vulnerable Populations during Service Disruptions. *Nat Hazards Rev* [Internet]. 2020 Nov;21(4):04020034. Available from: <http://ascelibrary.org/doi/10.1061/%28ASCE%29NH.1527-6996.0000401>